# PLUG-AND-PLAY RETRIEVAL-AUGMENTED ACTIVE TEST-TIME ADAPTATION FOR VLMS

## ABSTRACT

Pre-trained vision-language models (VLMs) have demonstrated remarkable performance across various real-world benchmarks. In particular, CLIP, one of the famous VLMs, has achieved satisfactory performance on vision-language tasks without fine-tuning (*i.e.,* zero-shot setting). Nevertheless, it is well-known that effectively leveraging a pre-trained model requires adaptation to the test distribution. Since the test distribution is typically unknown, test-time adaptation (TTA) has emerged as one of the solutions. However, existing TTA algorithms rely not on expert-provided ground-truth knowledge but on pseudo-labels derived from the knowledge of the pre-trained model itself. This undesirable reliance can lead to a cascade of incorrect knowledge propagation. To address this issue, we propose a novel framework, active test-time adaptation, which selectively queries human experts for ground-truth labels of uncertain samples and incorporates them for answering future queries. Then, we develop a novel algorithm, **RE**trieval-augmented **ACT**ive TTA (**REACT**), which is designed to be plug-and-play with any TTA algorithms. Through extensive experiments on ten real-world benchmarks commonly used in CLIP evaluation as well as a domain transfer benchmark based on ImageNet, the proposed algorithm is shown to effectively identify and query informative samples, leveraging them to enhance test-time inference capabilities.

## 1 INTRODUCTION

Vision-language models (VLMs) have received significant attention by integrating human language into various computer vision tasks. For example, various VLMs (Radford et al., 2021; Bai et al., 2023; Li et al., 2022; 2023; Liu et al., 2024; Bai et al., 2025) have been introduced and have demonstrated remarkable performance on zero-shot classification, captioning, image-text retrieval, and so on. In detail, CLIP (Radford et al., 2021) models, one of the pioneering VLMs, map text including class candidates and image inputs into a shared embedding space, respectively, and classify a given image by selecting the label with the highest matching score.

Recent studies (Feng et al., 2023b; Shu et al., 2022a; Yoon et al., 2024; Karmanov et al., 2024) began to propose the test-time adaptation strategies for VLMs, enabling them to quickly adjust to domain shifts in real-world scenarios. For example, TPT (Feng et al., 2023b) and DiffTPT (Feng et al., 2023a) optimize prompts for each test image via various visual augmentations to boost the confidence of the CLIP model's predictions. In contrast, the authors (Karmanov et al., 2024) proposed a backpropagation-free test-time adaptation algorithm called TDA that leverages a Tip-adapter (Zhang et al., 2022), which is a train-free adapter with a few samples, to reduce inference costs—a known drawback of previous

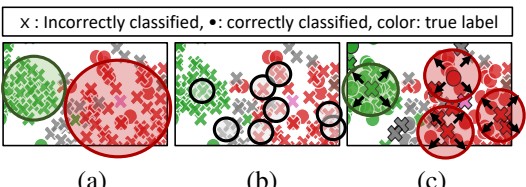

Figure 1: **Conventional TTA has difficulty classifying unseen classes. (a)** In this t-SNE plot of a conventional TTA, samples are correctly clustered by class (green, red) but are entirely misclassified. In contrast, our approach **REACT (b)** first selects informative samples as anchors for human labeling. **(c)** Subsequently, these labeled anchors propagate correct labels to nearby misclassified samples, improving overall accuracy.

prompt optimization techniques. However, as illustrated in Figure 1, these methods struggle to classify unseen classes using only the inherent knowledge of the pre-trained model. In particular, TDA

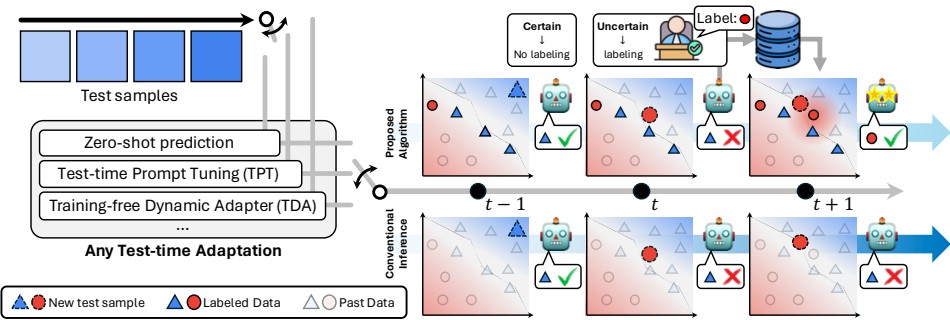

Figure 2: **Overall framework of REACT.** It identifies and labels uncertain samples from any test-time adaptation method. Future samples are then predicted by retrieving similar labeled examples. Even if **REACT** makes incorrect predictions (*i.e.,* at time $t$), it consults a labeler to refine the predictions and stores ground-truth labels in the database for later retrieval (*i.e.,* at time $t + 1$), ultimately improving classification performance through retrieval-augmented correction.

can correctly classify only a small fraction of the samples that are initially misclassified by CLIP. Our findings indicate that when a few labels from selected samples among the uncertain ones are provided by a labeler, the majority of these ambiguous cases can be correctly recovered. Further, several works (Lin et al., 2022; Pearl, 2009) have demonstrated that *it is theoretically infeasible to learn unseen classes without extra information*.

Regarding the critical limitation of TTA, simATTA (Gui et al., 2024) was the first work that emphasized the necessity of extra information for TTA and proposed a sample selection strategy to acquire and use labels for updating the model through lightweight training. However, this training-based approach accompanies high computational overhead and likely incurs catastrophic forgetting that undermines the knowledge from pre-training data. Therefore, we seek to investigate the feasibility of *training-free* TTA for VLMs, posing the following research question: *How can we design a train-free method that incorporates extra information into TTA while improving performance?*

In response to the question, our framework introduces a novel *retrieval-augmented* correction, which significantly enhances TTA without requiring additional model training. Our approach maintains and strategically leverages a growing database of labeled examples. Maintaining this database and appropriately retrieving labeled examples effectively substitute the training requirement. As illustrated in Figure 2, the test sample is confidently classified (*i.e., certain*) at time $t - 1$, so we simply rely on a conventional TTA method to make a prediction. In contrast, at time $t$, the model identifies a more challenging sample as *uncertain*, prompting us to query a labeler for its ground-truth label. Then, at time $t + 1$, we again encounter an uncertain sample. Differently from time $t$, by retrieving proper labeled examples, we can correct its label to the circle ($\circ$) based on the neighboring labeled example obtained at time $t$. This retrieval-augmented correction process allows us to make the correct classification, whereas existing TTA methods fail to do so.

Obviously, the key challenge for retrieval-augmented correction lies in minimizing the correction error. This challenge requires making strategic decisions about which samples need human labeling and which can be effectively corrected by retrieving labeled examples. Our approach—plug-and-play **RE**trieval-augmented **ACT**ive test-time adaptation (**REACT**)—addresses the challenge by carefully analyzing the relationships between test samples and existing labeled examples. We observe that the effectiveness of retrieval-augmented correction heavily depends on the quality of neighboring labeled examples. For example, if labeled examples in close proximity to a test sample consistently share the same label, the likelihood of making a correct prediction significantly increases. Motivated by this insight, we introduce two novel criteria to make optimal decisions at test time: *consistency* and *reliability*. The former guarantees high consistency with retrieved examples in terms of labels, and the latter guarantees high similarity to retrieved examples. These criteria enable us to selectively apply retrieval-augmented correction only when it is likely to succeed, thereby minimizing the correction error.

Our contributions are summarized as follows:

- We introduce a novel framework for active test-time adaptation of VLMs, **REACT**, which exploits retrieval-augmented correction with a labeled dataset. Importantly, owing to its training-free aspect, **REACT** can be incorporated into any conventional TTA methods.

- We propose a retrieval-augmented correction strategy for minimizing the errors by designing consistency and reliability from neighboring labeled examples.
- Extensive experiments on both out-of-distribution and cross-domain benchmarks demonstrate that **REACT** significantly enhances the performance of state-of-the-art TTA methods, paving the way for more robust and cost-efficient adaptation in real-world scenarios.

## 2 PRELIMINARY

### 2.1 TEST-TIME ADAPTATION WITH VLMS

CLIP (Radford et al., 2021), one of the famous VLMs, consists of two encoders: an image encoder $f_{\text{img}}$ and a text encoder $f_{\text{txt}}$. Its classification process with a class set $\mathcal{C}$ relies on the similarity score for a pair of an image and a text prompt. Given an image $x_t$, the similarity score is formulated as

$$s_c = \cos(f_{\text{img}}(x_t), f_{\text{txt}}(t_c)), \tag{1}$$

where $t_c$ is the text prompt for class $c \in \mathcal{C}$, and $\cos(\cdot, \cdot)$ is the cosine similarity. By using this similarity score, the prediction probability of each class is formulated as

$$z_t^c = \frac{\exp(s_c/\tau)}{\sum_{i=1}^{C} \exp(s_i/\tau)}, \tag{2}$$

where $\tau$ represents a temperature parameter. When applying a CLIP model to the TTA setup, we can unify the formulation of the prediction probability as

$$p(x_t) = \mathcal{T}_\xi(x_t; f_{\text{img}}, f_{\text{txt}}), \tag{3}$$

where $\mathcal{T}_\xi(\cdot)$ is the any TTA method with the pre-trained CLIP image encoder $f_{\text{img}}$ and text encoder $f_{\text{txt}}$, along with the context variables $\xi$ (*e.g.,* learnable prompts) that TTA requires (see Appendix A). However, such methods inevitably rely on pseduo-labels from the pre-trained model itself, thereby limiting their ability to handle severe distribution shifts or unseen classes. While we use CLIP as the VLM for simplicity, the same approach and its limitation apply equally to more advanced VLMs such as BLIP-2 (Li et al., 2023) and SigLIP (Zhai et al., 2023) (See Section D.1 for ablation studies).

### 2.2 PROBLEM FORMULATION

To overcome the limitation of pseudo-label-based TTA, we allow the model to occasionally query ground-truth labels for incoming test samples under a constrained budget. This idea is closely related to active test-time adaptation (ATTA) (Gui et al., 2024); once the label of a sample is acquired, it is immediately used to update the model, and the updated model can influence predictions for both future and even concurrent samples. We instead consider a more *realistic* problem formulation.

**Definition** (Our problem formulation). *For each test-time data sample $x_t \in \mathfrak{D}_{test}$ arriving at time $t$, the pre-trained VLM image encoder $f_{img}$ and text encoder $f_{txt}$ must decide whether to label it under the label budget $\mathfrak{B}$. Note that, due to real-time requirements in the TTA setup, the annotation for $x_t$ cannot be used as its prediction; instead, it is added to the labeled set $\mathfrak{D}_l(t)$ for use with future data.*

## 3 **REACT**: RETRIEVAL-AUGMENTED ACTIVE TEST-TIME ADAPTATION

The key challenges of our problem from Section 2.2 are (1) determining which samples should be labeled to better adapt to the test distribution and (2) effectively leveraging the labeled data for adaptation. Here is the brief summary of its process (See Appendix B for the overall design).

1. Apply *uncertainty-based filtering* to identify the samples that the model is most likely to misclassify. ▷ Section 3.1
2. Among uncertain samples, we construct $\mathfrak{D}_l(t)$ that is sufficiently populated (*i.e.,* larger than $N$) by adding labeled samples from an oracle. After constructing $\mathfrak{D}_l(t)$:
   (a) If the sample meets our criteria based on consistency and reliability, we perform *retrieval-augmented correction*. ▷ Section 3.2
   (b) Even if there is no remaining label budget, we still conduct retrieval-augmented correction upon *relaxed* criteria. ▷ Section 3.3
   (c) Otherwise, we query an oracle for the correct label and add the labeled sample into $\mathfrak{D}_l(t)$ for future use.

## 3.1 UNCERTAINTY-BASED FILTERING

We first choose the samples that the model is most likely to misclassify. For the test sample $x_t$, we first compute the logit $p(x_t) = \mathcal{T}_\xi(x_t; f_{\text{img}}, f_{\text{txt}})$ that is applied for the TTA method. Then, we calculate the entropy (Karmanov et al., 2024),

$$\mathcal{H}(p(x_t)) = -\sum_{c \in \mathcal{C}} p(y_t = c; x_t) \cdot \log p(y_t = c; x_t). \tag{4}$$

Since the entropy is a measure of *how uncertain a model is in its prediction for a given sample*, a sample with a high-entropy value is more likely to be predicted incorrectly by the model (Holub et al., 2008). Therefore, we conduct retrieval-augmented correction only for high-entropy samples.

## 3.2 RETRIEVAL-AUGMENTED CORRECTION

We employ our retrieval-augmented correction strategy that leverages $\mathfrak{D}_l(t)$ to correct the predictions made by TTA methods. However, blindly applying this correction to every sample can significantly increase correction errors. Thus, to selectively apply the correction, we propose two criteria: **consistency** for indicating whether the labeled samples nearest to a test sample share the same ground truth and **reliability** for measuring how close those labeled samples are to the test sample. Intuitively, if the nearest labeled samples have the same ground truth and are near the test sample, the test sample's ground truth is likely to match those labeled samples, reducing the chance of retrieval-augmented correction errors. For simplicity, we refer to the first and second nearest labeled samples as a *Referrer* $R = (x_t^R, y_t^R)$ and a *Verifier* $V = (x_t^V, y_t^V)$, respectively.

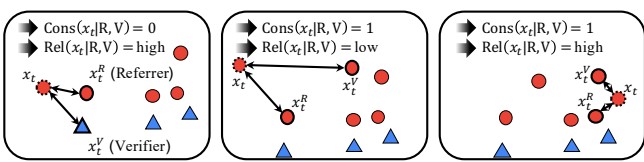

(a) Low consistency (b) High consistency, but low reliability (c) High consistency and high reliability

Figure 3: **Conceptual illustration of retrieval-augmented correction.** `REACT` operates under the philosophy of reflecting predictions based on proximity to a given test sample, provided the labels between neighboring samples are consistent.

Figure 3 depicts three common cases for the Referrer-Verifier interactions. In Figure 3a, the Referrer and Verifier disagree on the label, so no retrieval-augmented correction is applied. In Figure 3b, the labels match, but the Verifier is far from the test sample, making its endorsement of the Referrer's label less trustworthy. In contrast, Figure 3c shows both high consistency (same label) and high reliability (all close), thus allowing retrieval-augmented correction to proceed confidently. Based on these common cases, we now provide the detailed mathematical formulation as follows.

**Consistency.** To ensure that the candidate label from the Referrer is reliable, we define the *consistency* metric to check whether the Verifier's label agrees with the Referrer's. Formally, this metric is given by

$$\text{Cons}(x_t | R, V) = \mathbb{1}\{y_t^R = y_t^V\}, \tag{5}$$

where $\mathbb{1}\{\cdot\}$ is the indicator function. A value of 1 indicates full agreement between the Referrer and the Verifier, thus confirming the candidate label.

**Reliability.** To ensure that both the Referrer and the Verifier are close enough to a test sample $x_t$, we define the *reliability* to compute the average cosine similarity between $x_t$ and these two neighbors. Formally, this metric is given by

$$\text{Rel}(x_t | R, V) = \frac{1}{2}\Big(\cos\big(f_{\text{img}}(x_t), f_{\text{img}}(x_t^R)\big) + \cos\big(f_{\text{img}}(x_t), f_{\text{img}}(x_t^V)\big)\Big). \tag{6}$$

A higher reliability score indicates that both neighbors are strongly aligned with $x_t$, thereby increasing our confidence in the retrieval-augmented correction.

Overall, the condition for retrieval-augmented correction is thus formulated as

$$\mathcal{C}(x_t | R, V) = \text{Cons}(x_t | R, V) \cdot \text{Rel}(x_t | R, V). \tag{7}$$

This multiplicative formulation guarantees that if the Verifier disagrees with the Referrer (*i.e.,* , $\text{Cons}(x_t | R, V) = 0$), no retrieval-augmented correction is applied.

| Method | IN | IN-A | IN-V2 | IN-R | IN-S | Avg (All5) | Avg (OOD) |
|---|---|---|---|---|---|---|---|
| **ResNet-50** | | | | | | | |
| Zero-shot | 56.47 | 26.96 | 50.83 | 55.87 | 31.50 | 44.33 | 41.29 |
| +**REACT** | 57.66±0.06 | 28.11±0.19 | 50.76±0.07 | 56.93±0.07 | 36.94±0.04 | 46.08 | 43.18 |
| TPT | 60.68±0.01 | 25.12±0.07 | 54.33±0.23 | 59.11±0.04 | 35.29±0.04 | 46.91 | 43.46 |
| +**REACT** | 61.20±0.10 | 26.28±0.05 | 54.34±0.02 | 59.65±0.18 | 39.67±0.18 | 48.23 | 44.98 |
| C-TPT | 60.42±0.06 | 23.25±0.06 | 54.13±0.13 | 57.72±0.01 | 34.76±0.01 | 46.05 | 42.46 |
| +**REACT** | 61.10±0.11 | 24.76±0.30 | 53.88±0.03 | 58.66±0.12 | 39.62±0.02 | 47.60 | 44.23 |
| TDA[†] | 57.78±0.01 | 27.84±0.01 | 51.10±0.03 | 57.01±0.01 | 33.85±0.11 | 45.52 | 42.45 |
| +**REACT** | 58.64±0.01 | 28.73±0.28 | 51.31±0.09 | 57.84±0.06 | 39.00±0.01 | 47.10 | 44.22 |
| TDA | 61.72±0.04 | 30.59±0.21 | **55.10**±0.26 | 62.76±0.04 | 38.05±0.04 | 49.64 | 46.62 |
| +**REACT** | **62.06**±0.04 | **31.00**±0.15 | 54.99±0.29 | **63.25**±0.12 | **42.73**±0.10 | **50.80** | **47.99** |
| **ViT-B/16** | | | | | | | |
| Zero-shot | 63.52 | 54.65 | 59.80 | 75.36 | 44.05 | 59.48 | 58.47 |
| +**REACT** | 65.90±1.03 | 55.85±0.74 | 59.69±0.14 | 76.33±0.02 | 50.29±0.13 | 61.61 | 60.54 |
| TPT | 68.76±0.01 | 52.99±0.06 | 63.20±0.04 | 76.97±0.10 | 47.86±0.02 | 61.95 | 60.25 |
| +**REACT** | 69.27±0.12 | 54.89±0.05 | 61.94±1.49 | 77.18±0.08 | 51.30±0.94 | 62.91 | 61.32 |
| C-TPT | 68.31±0.01 | 50.71±0.04 | 62.50±0.04 | 75.68±0.08 | 47.46±0.06 | 60.93 | 59.08 |
| +**REACT** | 69.09±0.01 | 53.09±0.17 | 62.14±0.03 | 76.39±0.13 | 52.13±0.01 | 62.57 | 60.94 |
| TDA[†] | 65.81±0.81 | 55.42±0.06 | 60.06±0.13 | 75.99±0.13 | 46.01±0.08 | 60.66 | 59.37 |
| +**REACT** | 67.48±0.03 | 56.49±0.20 | 60.09±0.12 | 76.70±0.08 | 51.23±0.02 | 62.40 | 61.13 |
| TDA | 70.08±0.88 | 60.13±0.33 | 64.41±0.16 | 80.52±0.06 | 50.96±0.06 | 65.22 | 64.01 |
| +**REACT** | **70.88**±0.11 | **60.73**±0.60 | **64.54**±0.10 | **80.83**±0.11 | **55.01**±0.07 | **66.40** | **65.28** |

Table 1: **Results on the out-of-distribution benchmark.** All the compared methods are built upon CLIP-ResNet-50 or CLIP-ViT-B/16 baselines. The two average metrics *All 5* and *OOD* are calculated by taking the mean accuracy across all five datasets and four OOD datasets excluding ImageNet (IN). Note that the **bold** type represents the best performance overall.

## 3.3 CORRECTION AFTER LABEL BUDGET

As described in Section 3.2, we request samples for labeling if Equation 7 is below a certain threshold. As we continue labeling, we might eventually reach our labeling budget $\mathfrak{B}$ and no longer be able to request new labels from the oracle. Even in this scenario, we would like to benefit from retrieval-augmented correction. However, since proper labeled examples could not be inserted into the labeled dataset, there is a risk that the retrieval-augmented correction leveraging the Referrer and the Verifier may run incorrectly. To mitigate this issue, we choose the top-5 predictions from TTA as the Verifiers instead of the second nearest labeled sample and only proceed with corrections when matching the labels of the Verifiers and the Referrer. This technique is formulated as

$$p(x_t) = \begin{cases} \texttt{one\_hot}(y_t^R), & \text{if } y_t^R \in \text{Top\_5}\big(p(x_t)\big), \\ \mathcal{T}_\xi(x_t; f_{\text{img}}, f_{\text{txt}}), & \text{otherwise.} \end{cases} \tag{8}$$

Here, $\text{Top\_5}\big(p(x_t)\big)$ denotes the set of labels associated with the five highest confidence predictions generated by the TTA method. This technique enables our framework to apply corrections by setting the Referrer's label whenever it appears within these top-5 predictions (which function as the Verifiers), even in situations where no additional labeling resources are available. If the Referrer's label is not present in the top-5 predictions from TTA, we default to using the TTA's prediction instead.

## 4 EXPERIMENT

We provide details of the experimental setup, including datasets, baselines, and implementation details, in Appendix C due to space constraints. This section focuses on the main results and ablation studies using CLIP for the VLM. See Appendix D for additional results, particularly those involving more advanced VLMs such as BLIP-2 (Li et al., 2023) and SigLIP (Zhai et al., 2023).

---

† Since TDA uses a different prompt template than other baselines, we report its results with a unified template for fair comparison.

| Method | Aircraft | Caltech101 | Cars | DTD | EuroSAT | Flowers102 | Food101 | Pets | SUN397 | UCF101 | Average |
|---|---|---|---|---|---|---|---|---|---|---|---|
| **ResNet50** | | | | | | | | | | | |
| Zero-shot | 15.30 | 80.61 | 56.27 | 37.53 | 25.89 | 56.44 | 73.45 | 80.46 | 58.16 | 59.16 | 54.33 |
| +**REACT** | 18.16±0.25 | 84.61±0.26 | 57.54±0.42 | 43.35±1.05 | **74.16**±1.12 | 72.55±0.06 | 73.73±0.08 | 81.71±0.11 | 59.41±0.08 | 70.15±0.62 | 63.53 |
| TPT | 16.02±0.47 | 85.36±0.40 | 57.95±0.16 | 39.69±0.37 | 28.04±0.03 | 60.22±0.40 | 73.63±0.02 | 78.39±0.12 | 59.86±0.12 | 60.77±0.34 | 55.99 |
| +**REACT** | 18.60±0.25 | 87.73±0.83 | **60.31**±0.52 | 45.16±0.50 | 62.16±0.69 | 70.61±0.40 | 73.89±0.04 | 80.86±0.02 | 61.45±0.13 | 69.20±0.52 | 62.99 |
| C-TPT | 13.70±0.15 | 85.82±0.20 | 56.51±0.06 | 40.58±0.29 | 23.13±0.45 | 61.29±0.21 | 73.13±0.04 | 80.04±0.13 | 59.54±0.04 | 59.31±0.02 | 55.30 |
| +**REACT** | 16.65±0.51 | 87.59±0.69 | 59.35±0.11 | 45.89±0.96 | 60.76±0.78 | 73.02±1.23 | 73.64±0.03 | 81.85±0.38 | 61.28±0.45 | 69.83±0.25 | 62.98 |
| TDA[†] | 15.93±0.81 | 85.13±0.03 | 57.17±0.08 | 39.34±0.29 | 36.71±1.01 | 59.24±0.11 | 74.62±0.04 | 80.69±0.10 | 59.68±0.11 | 60.75±0.26 | 56.92 |
| +**REACT** | 17.75±0.74 | 85.52±0.23 | 58.48±0.09 | 44.18±0.79 | 70.81±1.08 | 71.64±0.66 | 74.63±0.19 | 81.84±0.06 | 60.40±0.04 | 68.86±0.98 | 63.41 |
| TDA | 16.79±0.23 | 89.36±0.43 | 57.36±0.17 | 43.91±0.08 | 41.66±0.61 | 68.17±0.28 | **77.78**±0.11 | 86.29±0.27 | 62.48±0.01 | 64.03±0.15 | 60.78 |
| +**REACT** | **19.01**±0.02 | 89.57±0.01 | 58.67±0.33 | **48.17**±0.08 | 71.08±1.29 | 77.19±0.63 | 77.28±0.20 | **86.62**±0.50 | **62.81**±0.11 | 71.27±1.12 | **66.16** |
| **ViT-B/16** | | | | | | | | | | | |
| Zero-shot | 22.59 | 85.76 | 65.61 | 40.60 | 44.25 | 64.07 | 82.68 | 83.81 | 63.68 | 66.30 | 61.94 |
| +**REACT** | 27.33±0.71 | 90.31±0.12 | 67.77±0.25 | 47.94±0.42 | **80.63**±0.54 | 80.53±0.20 | 83.57±0.06 | 87.10±0.13 | 65.30±0.03 | 76.28±0.54 | 70.67 |
| TPT | 23.43±0.08 | 93.21±1.07 | 66.52±0.20 | 46.33±1.10 | 42.69±0.16 | 68.25±0.93 | 83.94±1.18 | 84.42±0.98 | 65.19±0.47 | 67.62±0.41 | 64.16 |
| +**REACT** | 25.56±0.47 | 93.37±0.08 | 67.97±0.04 | 49.53±0.01 | 73.03±0.33 | 74.63±0.80 | 83.54±0.04 | 85.96±0.16 | 66.49±0.06 | 72.67±0.23 | 69.27 |
| C-TPT | 24.09±0.25 | 92.98±0.69 | 65.24±0.78 | 45.01±1.21 | 42.28±0.10 | 70.36±0.75 | 83.22±0.11 | 87.82±0.72 | 64.07±0.59 | 65.58±0.79 | 64.06 |
| +**REACT** | 27.78±0.47 | 93.27±0.18 | 66.88±0.37 | 50.59±0.17 | 74.72±0.79 | 77.49±0.43 | 83.73±0.05 | 87.70±0.02 | 66.22±0.05 | 74.65±0.64 | 70.30 |
| TDA[†] | 23.48±0.28 | 88.54±0.08 | 66.59±0.11 | 42.97±0.67 | 54.82±0.77 | 65.27±0.49 | 83.82±0.08 | 85.12±0.27 | 65.14±0.01 | 69.63±0.15 | 64.54 |
| +**REACT** | 25.74±0.21 | 90.41±0.43 | 67.94±0.04 | 49.41±0.34 | 75.40±0.54 | 79.20±0.49 | 84.20±0.06 | 87.37±0.10 | 66.18±0.01 | 76.01±0.62 | 70.18 |
| TDA | 25.29±0.30 | **94.06**±0.08 | 66.39±1.72 | 45.48±0.79 | 63.95±0.92 | 71.62±0.23 | **86.14**±0.06 | 89.85±0.13 | 67.70±0.04 | 70.94±0.28 | 68.14 |
| +**REACT** | 27.84±0.08 | **94.06**±0.03 | **68.76**±0.13 | **51.63**±0.88 | 79.61±1.40 | **82.28**±0.54 | 86.09±0.06 | **90.75**±0.18 | **68.20**±0.11 | **77.38**±1.01 | **72.66** |

Table 2: **Results on the cross-domain benchmark.** The *Average* is calculated by taking the mean accuracy across all ten datasets. Note that the **bold** type represents the best performance overall.

## 4.1 MAIN RESULTS

**Consistent performance gains with a plug-and-play design.** Our approach can be integrated into any test-time adaptation (TTA) pipeline, yielding substantial performance improvements. As shown in Table 1, when our plug-and-play module is added to a baseline TTA method, the top-1 accuracy on the OOD benchmark improves noticeably, ImageNet-A (65.8%→71.3%) and ImageNet-R (70.1%→75.8%), with similar gains observed on ImageNet-V2 and ImageNet-S, averaging around a 5% boost. On the cross-domain benchmark in Table 2, our method enhances performance as well, with datasets like Aircraft (82.0%→86.5%) and Food101 (73.5%→78.0%). These results demonstrate that our plug-and-play design consistently elevates performance across diverse datasets without the need for per-dataset tuning.

**Architecture-agnostic efficacy.** The effectiveness of our method is evident across different backbone architectures, demonstrating its architecture-agnostic nature. As shown in both the OOD and cross-domain benchmarks, integrating our module consistently improves performance regardless of whether the base model is ResNet-based or Transformer-based. For instance, for the OOD benchmark in Table 1, applying our method to TDA with a ResNet-50 backbone leads to an increase in average accuracy (49.64%→50.82%), while the ViT-B/16 backbone experiences a similar improvement (65.12%→66.49%). Likewise, for the Cross-Domain benchmark in Table 2, the ResNet-50 model benefits from a substantial accuracy gain (57.09%→63.29%), and the ViT-B/16 model achieves a comparable boost (68.08%→72.77%). These results highlight the adaptability of **REACT**, ensuring consistent performance gains across various architectures and datasets, making it a robust enhancement for both CNN- and Transformer-based vision models.

**Inference cost.** For a fair comparison, we run all experiments on Intel Xeon Gold 6326 CPUs with a single NVIDIA RTX 4090. We first warmed up the system with 100 samples and then measure the time taken to process 100 samples, as reported in Table 3. Our observations indicate that train-based TTA methods (*e.g.,* TPT and C-TPT) have an additional cost of roughly 500 ms per sample compared to train-free methods (*e.g.,* TDA). Notably, **REACT** only introduces a negligible overhead *below 1 ms* while still improving performance compared to each TTA method.

| **REACT** | Zero-shot | TPT | C-TPT | TDA |
|---|---|---|---|---|
| ✗ | 10.07 | 509.87 | 511.33 | 10.84 |
| ✓ | 10.16 | 510.22 | 511.55 | 11.27 |
| Rel. | +0.8% | +0.06% | +0.04% | +3.4% |

Table 3: **Per-sample cost analysis.** Our method adds only at most 0.4ms while consistently achieving improved accuracy when integrating with any TTA methods.

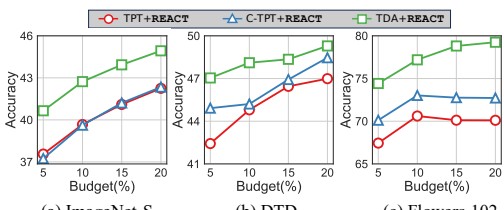

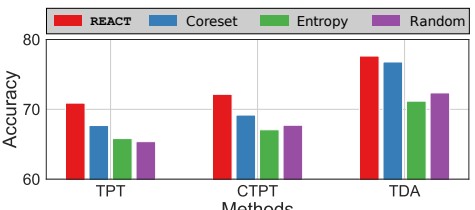

(a) ImageNet-S    (b) DTD    (c) Flowers-102

Figure 4: **Performance variation when controlling the budget across three datasets.** As the budget increases, the accuracy tends to rise owing to an enriched set of labeled examples.

Figure 5: **Comparison of conventional sample selection methods on Flowers-102.** We customize the active learning methods to be suitable for streaming data (see Appendix E).

### 4.2 FURTHER ANALYSIS RESULTS

**Larger budget makes more precise retrieval-augmented correction.** The retrieval-augmented approach relies on the labeled dataset $\mathfrak{D}_l(t)$. Allocating a larger budget enables the model to search through a broader range of potential matches, leading to more accurate identification of relevant examples and, in turn, higher final prediction accuracy. To validate this hypothesis, we conducted experiments under varying budget settings across three different datasets, and summarized the outcomes in Figure 4. The results consistently show that as the budget increases, accuracy improves for each dataset, highlighting the importance of dedicating sufficient resources for retrieval. Moreover, this trend remains robust across diverse datasets, underscoring the importance of reserving a sufficient budget.

**Comparison with diverse sample selection methods.** To enhance adaptation under shifting test distributions, **REACT** naturally involves a sample selection mechanism. We evaluate **REACT** against standard active learning methods, Coreset (Sener & Savarese, 2018), Entropy (Holub et al., 2008), and Random, on the Flowers-102 dataset, as presented in Figure 5. Since these active learning approaches are not originally designed for streaming data, we adapt them to an online setting (see Appendix E). Across all active learning methods, **REACT** consistently outperforms these baselines, illustrating its ability to select more informative samples for constructing the labeled dataset $\mathfrak{D}_l(t)$. This performance highlights the robustness of our approach in streaming environments and underscores the value of its sample selection mechanism.

**Hyperparameter sensitivity.** **REACT** introduces two hyperparameters, $\tau_u$ and $\tau_r$, that decide whether a sample should undergo retrieval-augmented correction. First, $\tau_u$ dictates how many samples rely on TTA predictions. Specifically, lowering $\tau_u$ makes more samples appear *uncertain*, diverting them away from direct TTA-based decisions, whereas raising $\tau_u$ increases the likelihood of accepting TTA outputs. Second, $\tau_r$ governs which of these uncertain samples undergo retrieval-augmented correction. A lower $\tau_r$ includes more samples in retrieval-augmented correction, while a higher $\tau_r$ imposes stricter conditions, causing more samples to label.

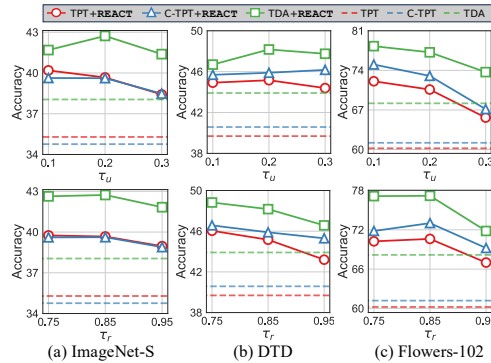

(a) ImageNet-S    (b) DTD    (c) Flowers-102

Figure 6: **Hyperparameter sensitivity.** We vary the uncertainty threshold $\tau_u$ with $\tau_r = 0.85$ (upper) and the retrieval-augmented correction threshold $\tau_r$ with $\tau_u = 0.2$ (lower).

To assess potential sensitivity of **REACT**, we tested different values of $\tau_u$ and $\tau_r$, as reported in Figure 6. Results across three datasets show that each one behaves differently due to varying levels of difficulty in each dataset. Nonetheless, on every dataset and for every TTA method, incorporating **REACT** consistently surpasses the baseline (*i.e., without* **REACT**), regardless of the specific hyperparameter settings. This observation confirms the robustness for a reasonable range of the thresholds.

**REACT selects the labeled samples in the boundary.** Figure 7 illustrates how **REACT** reshapes the decision boundary. Here, the examples with black outlines highlight the samples that **REACT** selected for labeling, which act as a Referrer or a Verifier. They have a noticeable influence on their

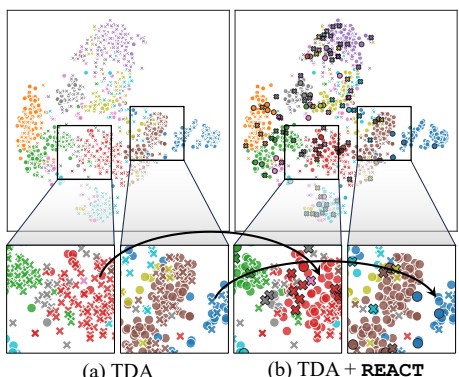

(a) TDA      (b) TDA + **REACT**

Figure 7: **Partial t-SNE (Van der Maaten & Hinton, 2008) visualization to check the decision boundary changes (left) and correct misalignment of image and text (right) by retrieval-augmented correction as the labeled dataset $\mathfrak{D}_l(t)$ is built on EuroSAT.** Circles (∘) represent samples where the prediction matches the ground truth, while crosses (×) indicate samples that were misclassified. Black outlines indicate the samples selected and labeled through **REACT**.

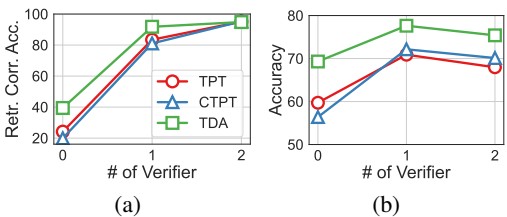

(a)      (b)

Figure 8: **Performance changes as the number of Verifiers increases.** (a) indicates the accuracy specifically for retrieval-augmented correction, whereas (b) refers to the accuracy calculated over all samples.

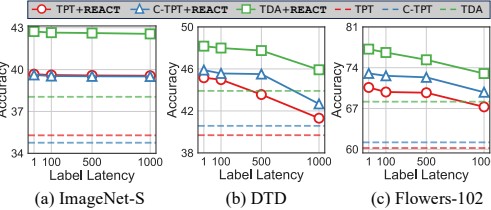

(a) ImageNet-S    (b) DTD    (c) Flowers-102

Figure 9: **Robustness for label latency in REACT.** Note that *labeling latency* is defined as the number of samples between when **REACT** decides to label a sample and when it is labeled.

neighborhood where many samples initially misclassified (depicted as crosses in Figure 7a) have their predictions corrected and align with the ground truth (depicted as circles in Figure 7b). This visualization clearly demonstrates the power of strategically chosen labeled samples in guiding the overall prediction landscape.

**REACT corrects the sample prediction that misaligns image and text embeddings.** Figure 7b shows that **REACT** can use a set of carefully selected labeled examples to properly correct false predictions stemming from misalignment issue in TTA methods. By leveraging retrieval-augmented correction, **REACT** demonstrates its ability to correct sample prediction that misaligns image and text embeddings in TTA methods. Accordingly, this finding verifies the fuel-efficiency of **REACT** that works with a small amount of labeled data (∼ 10% of the test set).

**Impact of a Verifier** $(x_t^V, y_t^V)$**.** The role of a Verifier is to minimize errors in retrieval-augmented correction, as argued in Section 3.2. To validate this hypothesis, we conducted experiments by varying the number of Verifiers and summarized the results in Figure 8. As shown in Figure 7a, introducing the Verifier concept (*i.e.,* $0 \rightarrow 1$) leads to a significant boost in retrieval-augmented correction accuracy. Additionally, the retrieval-augmented correction accuracy increases as the number of Verifiers increases. On the other hand, Figure 7b demonstrates that the overall accuracy is dropped when the number of Verifiers increases (*i.e.,* $1 \rightarrow 2$). This degradation is attributed to Equation 7 becoming overly strict as the number of Verifiers increases, which in turn reduces the number of samples where retrieval-augmented correction is applied.

**Sensitivity of labeling latency of REACT.** Although we used the ground truth as the answers from an oracle, because a powerful VLM is likely to act as an oracle in real-world applications, some delays in labeling are unavoidable. Figure 9 shows that even though longer delays can gradually reduce the performance gap of **REACT** over traditional TTA methods, **REACT** still maintains robustness by at least matching TTA's performance. We evaluated this sensitivity on three datasets, ImageNet-S (50,000 samples), DTD (about 2,000 samples), and Flowers-102 (about 2,000 samples), using labeling delays of 1, 100, 500, and 1,000 samples. On the large ImageNet-S dataset, the performance remained stable at around 43% for TDA+**REACT** and 40% for C-TPT+**REACT**. However, on the smaller DTD and Flowers-102 datasets, the same delays led to more noticeable drops in accuracy (*e.g.,* DTD: 46%→ 42% and Flowers-102: 74%→67%). These results confirm that performance degradation occurs when the labeling latency constitutes a significant portion of the entire test set size. In other words, increasing the size of the streaming set is expected to minimize the negative impact on performance.

## 5 RELATED WORK

**Vision language models (VLMs).** To comprehend the visual and language representations, multiple approaches have been explored (Lu et al., 2019; Das et al., 2017; De Vries et al., 2017; Qi et al., 2020; Gan et al., 2020; Yu et al., 2021; Li et al., 2020). In the stream of trials to understand both modalities at once, CLIP (Radford et al., 2021) emerged in 2021, drawing significant attention due to its remarkable zero-shot performance across various tasks. In a similar vein, ALIGN (Jia et al., 2021) was introduced, employing a comparable training methodology but featuring distinct architectural and training dataset characteristics. Unlike CLIP, BLIP (Li et al., 2022) introduced a captioning module aimed at improving model performance by rectifying noisy captions. LiT (Zhai et al., 2022) and BLIP-2 (Li et al., 2023) enhanced training efficiency by freezing specific encoder parameters. FILIP (Yao et al., 2021) endeavored to enable a model to discern finer image details through a fine-grained, *i.e.,* patch-level, training approach.

**Test-time adaptation for VLMs.** When transferring the zero-shot capabilities of VLMs, the distribution shift between pre-training data and test data is the main obstacle. Test-time adaptation (TTA) methods for VLMs have been proposed to adapt VLMs to the distribution shift. These methods adapt to an input test image on-the-fly, without any training requirements; they usually leverage the output of a VLM because only a single unlabeled test image is available. TPT (Shu et al., 2022b) uses data augmentation to enrich the test image, filters out unreliable augmented images based on prediction entropy, and then updates learnable prompts by minimizing the entropy of the reliable predictions. DiffTPT (Feng et al., 2023a) enhances TPT by augmenting an input image with informative and diverse images generated from a pre-trained diffusion model. C-TPT (Yoon et al., 2024) also enhances TPT by calibrating the prediction uncertainty. Unlike previous methods, TDA (Karmanov et al., 2024) introduces training-free TTA by leveraging Tip-Adapter (Zhang et al., 2022), thereby boosting performance while lowering inference cost. However, existing methods solely rely on the *internal* knowledge encoded in the VLM parameters which are constrained to the pre-training data (Agarwal et al., 2021), as opposed to our framework featuring retrieval-augmented correction.

**Active learning.** Active learning (Settles, 2009; Ren et al., 2021; Geifman & El-Yaniv, 2019; Munjal et al., 2022) aims to minimize human labeling costs by identifying informative data that maximize model performance. Research in this area generally follows two main trajectories: uncertainty-based sampling and diversity-based sampling. In the former, prediction probability-based strategies such as soft-max confidence (Lewis & Catlett, 1994), margin (Roth & Small, 2006), and entropy (Holub et al., 2008) are straightforward yet effective approaches. In the latter, diversity-based strategies (Sener & Savarese, 2018; Parvaneh et al., 2022) employ clustering or coreset selection protocols, such as the coreset method (Sener & Savarese, 2018) which aims to maximize coverage distance across unlabeled data. Additionally, hybrid methods like BADGE (Ash et al., 2019) combine uncertainty and diversity by performing $k$-means++ clustering in gradient embedding space. More recently, PCB (Bang et al., 2024) firstly proposed the active learning framework for VLMs, which focused on the balance of classes using pseudo labels when selecting samples. While most active learning research focuses on static datasets, a few studies (Qin et al., 2021; DeSalvo et al., 2021) investigate online active learning in data streams. However, they do not account for shifts in the data stream, such as out-of-distribution or cross-domain variations. simATTA (Gui et al., 2024) is the first algorithm developed for *active test-time adaptation*, but it necessitates training of the model, which is susceptible to catastrophic forgetting and expensive.

## 6 CONCLUSION

We have proposed **REACT**, a framework that integrates active sample selection with retrieval-augmented correction to boost pre-trained VLMs under distribution shifts. By querying uncertain samples for labeling, **REACT** builds a compact yet informative labeled dataset. During inference, it effectively corrects uncertain predictions by exploiting this extra information. Experiments on OOD and cross-domain benchmarks show that **REACT** boosts accuracy by up to 7.68 percentage points across two backbones (ResNet and ViT) with minimal overhead, offering an efficient plug-and-play solution for TTA under limited label budgets.

## ETHICAL CONSIDERATIONS

This work focuses on the development of a general-purpose algorithm, and no direct ethical issues arise from the research process itself. Moreover, all experiments were conducted exclusively using widely-used public academic benchmark datasets, such as ImageNet and Caltech101, thereby no new data containing personally identifiable or sensitive information was collected, processed, or distributed. We therefore conclude that our work does not raise any major ethical issues.

## REPRODUCIBILITY STATEMENT

To ensure reproducibility, we provide comprehensive implementation details including the `REACT` algorithm (See Algorithm 1), mathematical formulations for all components (See Section 3), and fixed hyperparameters. Experimental setup details are described in Appendix C, covering all 15 benchmark datasets, baseline implementations, and evaluation metrics. Source code is available at `https://anonymous.4open.science/r/react_iclr26` with implementations for all TTA baselines and evaluation scripts for reproducibility.

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

-Supplementary Material-

# Plug-and-Play Retrieval-Augmented Active Test-Time Adaptation for VLMs

## A    CONVENTIONAL TTA

fTest-time adaptation (TTA) using vision-language models (VLMs) can be classified into two types: prompt tuning-based approaches (*e.g.,* TPT (Shu et al., 2022b) and C-TPT (Yoon et al., 2024)) and training-free approaches (*e.g.,* TDA (Karmanov et al., 2024)). Depending on the approach, we can formulate the TTA operator $\mathcal{T}_\xi(\cdot)$ as follows. Note that C-TPT is omitted here due to its conceptual similarity to TPT.

### A.1    TEST-TIME PROMPT TUNING (TPT)

TPT is achieved by fine-tuning a learnable prompt $p$ using a single test sample $x_t$. Given multiple augmented views of $x_t$, denoted by $A_i(x_t)$ for $i = \{1, \ldots, M\}$, the goal is to minimize the entropy (as defined in Equation 4) of the model's predictions across these augmentations.

**Objective Function.**   The probability distribution over classes generated by the CLIP model with prompt $p$ on the $i$-th augmented view can be formulated as

$$p^{(i)}(x_t) = p(A_i(x_t)) = \frac{\exp(\mathbf{s}^{(i)}/\tau)}{\sum_{k=1}^{\mathcal{C}} \exp(s_k^{(i)}/\tau)},$$

where $\mathbf{s}^{(i)} = \{s_c^{(i)} | c \in \{1, \ldots, \mathcal{C}\}\}$, and the similarity $s_c^{(i)}$ is defined as

$$s_c^{(i)} = \cos\left(f_{\mathrm{img}}(A_i(x_t)), f_{\mathrm{txt}}([p; t_c])\right),$$

where $t_c$ is the text prompt of class $c$. Note that it is similarly calculated by Equation 3, but the difference is that the text prompt concatenates learnable prompt $p$ and text prompt $t_c$ for class $c$, and the input image is augmented when calculating the similarity score. After then, the averaged prediction $\tilde{p}_p(x_t)$ (with confidence selection) is defined as

$$\tilde{p}_p(x_t) = \frac{1}{\rho N} \sum_{i=1}^{M} \mathbf{1}\left[\mathcal{H}(p^{(i)}(x_t)) \leq \tau\right] p^{(i)}(x_t),$$

where $\mathcal{H}(\cdot)$ denotes the entropy function, $\tau$ is the entropy threshold, and $\rho$ is the proportion of augmented views selected.

The optimization problem is then given by

$$p^* = \arg\min_p \mathcal{L}_{\mathrm{TPT}}(p; x_t),$$

with the loss defined as the marginal entropy,

$$\mathcal{L}_{\mathrm{TPT}}(p; x_t) = -\sum_{y \in \mathcal{Y}} \tilde{p}_p(x_t) \log \tilde{p}_p(x_t).$$

After optimization, the test-time adapted prediction function is defined as

$$\mathcal{T}_\xi(x_t) = \mathcal{T}_{p^*}(x_t) \triangleq p_{p^*}(x_t).$$

### A.2    TRAINING-FREE DYNAMIC ADAPTER (TDA)

In contrast to prompt tuning-based approaches, TDA leverages a non-parametric dynamic cache to adapt predictions without backpropagation. Here, $\xi$ comprises the components of the dynamic cache,

$$\xi = \{Q_p, \hat{L}_p, Q_n, \hat{L}_n\},$$

where $(Q_p, \hat{L}_p)$ denotes the keys and corresponding pseudo labels in the *positive cache* (collected from high-confidence predictions), and $(Q_n, \hat{L}_n)$ denotes the keys and corresponding negative pseudo labels in the *negative cache* (collected from low-confidence predictions).

Then, the final prediction is computed as

$$\mathcal{T}_\xi(x_t) = f_{\text{img}}(x_t)W_T^c$$
$$+ \mathcal{A}\big(f_{\text{img}}(x_t)Q_p^T\big)\,\hat{L}_p - \mathcal{A}\big(f_{\text{img}}(x_t)Q_n^T\big)\,\hat{L}_n,$$

where $W_T^c = \big[f_{\text{txt}}(t_c) \mid c \in \mathcal{C}\big]$ represents the text embedding matrix computed from the class names, and $\mathcal{A}(\cdot)$ is an adapter function (*e.g.,* an exponential weighting function as used in Tip-Adapter (Zhang et al., 2022)).

# B ALGORITHM DETAILS

Algorithm 1 outlines our proposed method, **REACT**. For each test sample $x_t$, our method first assesses its prediction uncertainty. If the sample is certain, **REACT** directly returns the base TTA prediction. For uncertain samples, it employs a hierarchical strategy to manage its labeling budget efficiently. The process begins with a brief warm-up phase, building a foundational labeled set $\mathfrak{D}_l$ by querying the oracle for the first $N$ uncertain samples encountered.

Once this foundational set is established, **REACT**'s core **retrieval-augmented correction** mechanism is activated for subsequent uncertain samples. It first attempts a budget-free correction by retrieving high-confidence matches from the existing labeled set $\mathfrak{D}_l$. Only if this retrieval fails to yield a confident correction and the labeling budget $\mathfrak{B}$ is not exhausted, the algorithm does query the oracle to annotate the sample. This strategy ensures that the valuable labeling budget is reserved for the most genuinely ambiguous and informative samples, making the adaptation process both effective and efficient.

---

**Algorithm 1: REACT**

---

**Input:** Labeled dataset $\mathfrak{D}_l(t)$, Uncertainty threshold $\tau_u$, Retrieval-augmented correction
threshold $\tau_r$, Labeling budget $\mathfrak{B}$, Minimum labeled samples $N$, TTA method $\mathcal{T}_\xi(\cdot)$,
Oracle labeler $\texttt{Oracle}(\cdot)$, Test sample $x_t$

Compute uncertainty $u \leftarrow \mathcal{H}\big(p(x_t)\big)$   ▷ Eq. (4)
\# **Certain sample**: Use TTA prediction   ▷ Section 3.1
**if** $u < \tau_u$ **then**
  |   **return** $p(x_t) = \mathcal{T}_\xi(x_t; f_{\text{img}}, f_{\text{txt}})$
\# **Uncertain sample**: Calculate criteria
**else**
  |   \# **Warm-up**: Construct $\mathfrak{D}_l(t)$
  |   **if** $|\mathfrak{D}_l(t)| < N$ **then**
  |     |   $\mathfrak{D}_l(t+1) \leftarrow \mathfrak{D}_l(t) \cup \{(x_t, \texttt{Oracle}(x_t))\}$
  |     |   **return** $p(x_t) = \mathcal{T}_\xi(x_t; f_{\text{img}}, f_{\text{txt}})$
  |   Retrieve R, V $\in \mathfrak{D}_l(t)$
  |   Compute $\mathcal{C}(x_t | \text{R}, \text{V})$   ▷ Eq. (7)
  |   \# Retrieval-augmented correction   ▷ Section 3.2
  |   **if** $\mathcal{C}(x_t | \text{R}, \text{V}) > \tau_r$ **then**
  |     |   **return** $p(x_t) = \texttt{one\_hot}(y_t^R)$
  |   \# Correction after label budget   ▷ Section 3.3
  |   **else if** $|\mathfrak{D}_l(t)| \geq \mathfrak{B}$ **then**
  |     |   Calculate the logit $p(x_t)$   ▷Eq. (8)
  |     |   **return** $p(x_t)$
  |   \# Annotate $x_t$ by oracle, and update $\mathfrak{D}_l(t+1)$
  |   **else**
  |     |   $\mathfrak{D}_l(t+1) \leftarrow \mathfrak{D}_l(t) \cup \{(x_t, \texttt{Oracle}(x_t))\}$
  |     |   **return** $p(x_t) = \mathcal{T}_\xi(x_t; f_{\text{img}}, f_{\text{txt}})$

---

## C Experimental Setup

### C.1 Datasets

Our experiments were conducted on two benchmarks: the out-of-distribution (OOD) benchmark and the cross-domain benchmark, both used in previous research on adapting VLMs during test time. The OOD benchmark evaluates the robustness of our approach by assessing its performance on four datasets derived from ImageNet (Deng et al., 2009), ImageNet-A (Hendrycks et al., 2021b), ImageNet-V2 (Recht et al., 2019), ImageNet-R (Hendrycks et al., 2021a), and ImageNet-S (Gao et al., 2022), which are specifically designed to test a model's ability to generalize to new and unseen data. In contrast, the cross-domain benchmark examines the model's adaptability across different domains by evaluating it on ten diverse image classification datasets, each representing a distinct class space. These datasets include Aircraft (Maji et al., 2013), Caltech101 (Fei-Fei et al., 2004), Stanford Cars (Krause et al., 2013), DTD (Cimpoi et al., 2014), EuroSAT (Helber et al., 2019), Flower102 (Nilsback & Zisserman, 2008), Food101 (Bossard et al., 2014), Oxford Pets (Parkhi et al., 2012), SUN397 (Xiao et al., 2010), and UCF101 (Soomro et al., 2012). Using these two benchmarks, we provide a comprehensive evaluation of the model's generalization capability and adaptability during test time. We summarize the statistics of datasets in Table 4, and belows are the details of each dataset.

#### C.1.1 OOD Benchmarks

**ImageNet** (Deng et al., 2009) contains 14,197,122 annotated images according to the WordNet hierarchy. This dataset has been used in the ImageNet large scale visual recognition challenge (ILSVRC), a benchmark in image classification and object detection, since 2010.

**ImageNet-A** (Hendrycks et al., 2021b) is a subset of 7,500 visually similar but naturally perturbed ImageNet images of 200 classes.

**ImageNet-V2** (Recht et al., 2019) consists of 10,000 images and 1,000 ImageNet classes, and was collected by applying an updated natural data collection pipeline to the original ImageNet dataset.

**ImageNet-R** (Hendrycks et al., 2021a) includes 30,000 images belonging to 200 categories of the ImageNet dataset, but with diverse artistic styles.

**ImageNet-S** (Gao et al., 2022) consists of 50,000 sketches of 1,000 class objects from the ImageNet dataset, and represents a domain shift from natural images to sketches.

#### C.1.2 Cross Domain Benchmark

**FGVC-Aircraft** (Maji et al., 2013) encompasses a total of 10,200 images depicting various aircraft. This dataset is organized into 102 distinct classes, and each class corresponds to a specific aircraft model variant. Notably, there are 100 images available for each of these 102 different aircraft model variants. The class name in this dataset is composed of the make, model, and specific variant, *e.g.,* Boeing 737-76J.

**Caltech101** (Fei-Fei et al., 2004) is composed of 101 unique object categories, each corresponding to a different type of objects or scenes. These categories encompass a wide range of objects, such as various animals, vehicles, and more. The dataset comprises a total of 9,000 images with varying numbers of images allocated to each category. Notably, it is considered a severely imbalanced dataset due to the uneven distribution of images across its categories.

**Stanford Cars** (Krause et al., 2013) consists of a collection of 16,185 images categorized into 196 different classes, with each class typically representing a specific car make, model, and year, *e.g.,* 2012 Tesla Model S.

**DTD** (Cimpoi et al., 2014), abbreviated from Describable Texture Dataset, is designed for texture classification task. This dataset consists of 47 distinct classes, including categories like fabrics and natural materials. In total, DTD comprises 5,640 samples. Notably, when examining the performance reported in the CLIP (Radford et al., 2021), it becomes evident that DTD poses a challenging problem for pre-trained CLIP models, as textures are not typical, easily recognizable objects.

| Dataset | # of Classes | # of Test Instances |
|---|---|---|
| ImageNet | 1000 | 50,000 |
| ImageNet-A | 200 | 7,500 |
| ImageNetV2 | 1000 | 10,000 |
| ImageNet-R | 200 | 30,000 |
| ImageNet-S | 1000 | 50,889 |
| FGVC Aircraft | 100 | 3,333 |
| Caltech101 | 100 | 2,465 |
| Stanford Cars | 196 | 8,041 |
| DTD | 47 | 1,692 |
| EuroSAT | 10 | 8,100 |
| Flowers102 | 102 | 2,463 |
| Food101 | 101 | 30,300 |
| Oxford Pets | 37 | 3,669 |
| SUN397 | 397 | 19,850 |
| UCF101 | 101 | 3,783 |

Table 4: **Profiles of the datasets used for the experiments.**

**EuroSAT** (Helber et al., 2019) comprises 10 distinct classes that represent various land use and land cover categories. In total, this dataset includes 27,000 satellite images, with 2,700 images allocated to each of the 10 classes. Notably, each class contains an equal number of images, ensuring a balanced distribution within the dataset.

**Flowers102** (Nilsback & Zisserman, 2008) consists of 102 different categories of flowers, each representing a distinct flower species such as roses, sunflowers, and daisies. There are 8,189 image and label pairs in total. Some categories have more images than the others, which means that it is imbalanced as typical real-world datasets; each category contains at least 40 and at most 258 samples.

**Food-101** (Bossard et al., 2014) consists of 101 food categories with 750 training and 250 test images per category, summing up to 101K images. The labels for the test images have been manually cleaned, while the training set contains some noise.

**Oxford Pets** (Parkhi et al., 2012) consists of 37 different pet categories, including various dogs and cats. This dataset contains 7,400 samples. In particular, it has 4,978 dog images and 2,371 cat images. We use only class labels even though the dataset has segmentation, *i.e.,* both RoI and class.

**SUN397** (Xiao et al., 2010) is the the Scene UNderstanding (SUN) database that contains 899 categories and 130,519 images. There are 397 well-sampled categories to evaluate numerous state-of-the-art algorithms for scene recognition.

**UCF101** (Soomro et al., 2012) is an extension of UCF50 and consists of 13,320 video clips, which are classified into 101 categories. These 101 categories can be classified into 5 types (body motion, human-human interactions, human-object interactions, playing musical instruments, and sports). The total length of these video clips is over 27 hours. All the videos are collected from YouTube and have a fixed frame rate of 25 FPS with the resolution of 320×240. In this work, the middle frame of each video is fed to the image encoder.

### C.2 BASELINES

We compare **REACT** against three state-of-the-art TTA methods for VLMs, which can be categorized into prompt-tuning based (*e.g.,* TPT, C-TPT) and training-free approaches (*e.g.,* TDA).

**Prompt Tuning-based TTA.** We select TPT (Shu et al., 2022b) and its concurrent extension C-TPT (Yoon et al., 2024) as the representatives of this category. These methods adapt a set of learnable prompt vectors for each individual test sample. The optimization is guided by a self-supervised objective, which aims to minimize the entropy of the model's predictions across multiple augmented views of that sample, thereby fine-tuning the prompt without requiring true labels.

**Training-free TTA.** As a training-free counterpart, we use TDA (Karmanov et al., 2024). In contrast to prompt tuning, TDA is a non-parametric method that requires no backpropagation at test time. It maintains a dynamic cache that stores features from past predictions, separating them into positive

| Model | Aircraft | Caltech101 | Cars | DTD | EuroSAT | IN | IN-A |
|---|---|---|---|---|---|---|---|
| CLIP | 22.59 | 85.76 | 65.61 | 40.60 | 44.25 | 63.52 | 54.65 |
| + **REACT** | **27.33** | **90.31** | **67.77** | **47.94** | **80.63** | **65.90** | **55.85** |
| BLIP-2 | 12.75 | 92.58 | 78.54 | 53.49 | 48.88 | 60.23 | 63.35 |
| + **REACT** | **22.20** | **92.60** | **79.92** | **53.66** | **56.58** | **61.83** | **63.89** |
| SigLIP | 36.66 | 95.86 | 88.47 | 61.70 | 33.88 | 75.14 | 43.49 |
| + **REACT** | **39.48** | **96.06** | **88.50** | **63.30** | **34.30** | **75.22** | **44.24** |

Table 5: **Performance with more recent VLMs.**

| Method | Aircraft | Caltech101 | Cars | DTD | EuroSAT | IN | IN-A |
|---|---|---|---|---|---|---|---|
| CLIP (Zero-shot) | 22.59 | 85.76 | 65.61 | 40.60 | 44.25 | 63.52 | 54.65 |
| CLIP + CoOp (Prompt learning) | 26.22 | 89.17 | 66.68 | 42.61 | 72.43 | **68.50** | 52.47 |
| CLIP+**REACT** (Train-free) | **27.33** | **90.31** | **67.77** | **47.94** | **80.63** | 65.90 | **55.85** |

Table 6: **Performance comparison of training-based TTA.**

and negative caches using confidence scores. These caches are then used to adjust the prediction scores of the current test sample, effectively adapting the model on the fly.

### C.3 IMPLEMENTATION DETAILS

All models in our experiments are built upon the pre-trained CLIP model, which consists of an image encoder and a text encoder. The image encoder can be either a ResNet or a Vision Transformer (ViT), while the text encoder is a Transformer. TTA is performed in a single-image setting with a batch size of 1. Specifically, the thresholds of uncertainty (Equation 4) and retrieval-augmented correction (Equation 7) are set to 0.2 and 0.85, respectively, the initial number of labeled examples $N$ is set to 5, and the label budget $\mathfrak{B}$ is set to 10% of the size of the test set. These hyperparameters remain fixed and are evaluated across various datasets. For evaluation, we use the *top-1 accuracy* (%) as the standard classification metric. All experiments were conducted three times on a single NVIDIA RTX 4090, and the results were then averaged with standard deviation.

## D FURTHER ANALYSES

### D.1 VARIOUS VLMS

While recent vision-language models (VLMs) such as BLIP-2 (Li et al., 2023) and SigLIP (Zhai et al., 2023) have emerged, outperforming traditional models like CLIP, the majority of prior TTA research has been conducted on CLIP. Therefore, for baseline consistency and fair comparison, we initially adopted CLIP in our main experiments (as presented in Table 1 and Table 2). To further showcase the generalizability of our approach, we extend our evaluation to include BLIP-2 and SigLIP. As shown in Table 5, applying **REACT** yields consistent improvements across these modern architectures. These findings underscore that **REACT** is a versatile and effective method, capable of enhancing a wide range of VLMs.

### D.2 COMPARISON WITH SUPERVISED TTA

To rigorously evaluate our method in an active TTA setup, we compare it against a supervised, training-based approach, CoOp (Zhou et al., 2022). For a fair comparison, both CoOp and CLIP+**REACT** utilize the same labeled set generated by **REACT**, and we summarize the results in Table 6. Critically, CoOp leverages this set for supervised fine-tuning with gradient updates. Despite this advantage of learning directly from labels, our training-free CLIP+**REACT** demonstrates superior performance across the majority of datasets. This outcome underscores a critical limitation of supervised TTA: even with access to labels, the effectiveness of training-based methods is fundamentally hampered by the computational constraints of the TTA setting, which permit only a few update iterations. These results confirm that CLIP+**REACT** is not only more efficient by sidestepping backpropagation but also achieves a more powerful adaptation, ultimately surpassing a supervised method under practical TTA constraints.

| Metric | Aircraft | Caltech101 | Cars | DTD | EuroSAT | IN | IN-A |
|---|---|---|---|---|---|---|---|
| Max Softmax | 27.69 | 85.76 | 59.25 | 48.05 | **82.14** | 64.54 | 55.89 |
| Energy | 25.74 | 94.04 | 67.57 | 44.74 | 61.07 | 70.70 | 60.42 |
| **Entropy (Ours)** | **27.84** | **94.06** | **68.76** | **51.63** | 79.61 | **70.88** | **60.73** |

Table 7: **Performance with various uncertainty metrics.**

## D.3 ABLATION ON UNCERTAINTY METRIC

To validate our choice of *entropy* as the uncertainty metric, we performed an ablation study by replacing it with two other prominent metrics: Max Softmax probability and Energy Score (Liu et al., 2020). In Table 7, the results clearly indicate that entropy is the most effective choice, achieving the highest accuracy on six out of the seven benchmark datasets. The performance gain is particularly notable on DTD, where entropy surpasses the next best metric by a significant margin. While Max Softmax showed a stronger result on the EuroSAT, the consistent top-tier performance of entropy across a diverse range of benchmarks underscores its general robustness.

## E CONVENTIONAL SAMPLE SELECTION

To evaluate the effectiveness of `REACT`, Figure 5 compared it against three well-known sample selection methods: Random, Coreset, and Entropy. We adapt each method for online use by incorporating a criterion threshold and tailoring Equation 7 to fit their specific approaches. Below, we describe each method and its customized criterion.

**Random.** This baseline randomly generates a value $v$ between 0 and 1. Retrieval-augmented prediction is triggered when $v$ exceeds the retrieval threshold $\tau_r$. The criterion is simple and defined as $\mathcal{C}(x_t) = v$.

**Coreset (Sener & Savarese, 2018).** This method focuses on selecting diverse samples based on model embeddings. It triggers annotation when the cosine similarity between a test sample $x_t$ and its nearest labeled sample $x_t^R$ (Referrer) falls below the retrieval threshold $\tau_r$. The criterion is formulated as

$$\mathcal{C}(x_t) = \cos\left(f_{\text{img}}(x_t), f_{\text{img}}(x_t^R)\right).$$

**Entropy (Holub et al., 2008).** This method selects samples with high uncertainty, measured using the entropy from the model's output probabilities (see Equation 4). Retrieval-augmented prediction is activated when the entropy $\mathcal{H}(p(x_t))$ exceeds the retrieval threshold $\tau_r$. Since we already apply uncertainty-based filtering with $\tau_u = 0.2$ (as noted in Section 3.1), we ensure $\tau_r > 0.2$. The criterion is formulated as

$$\mathcal{C}(x_t) = \mathcal{H}(p(x_t)) = -\sum_{c \in \mathcal{C}} p(y_t = c|x_t) \cdot \log p(y_t = c|x_t).$$

