# OpenReview forum: "Plug-and-Play Retrieval-Augmented Active Test-Time Adaptation for VLMs"
_ICLR.cc/2026/Conference — ICLR 2026 Conference Withdrawn Submission_

### Official Review · Reviewer_qyM8 · 2025-10-26

**Soundness:** 2
**Presentation:** 3
**Contribution:** 2
**Rating:** 2
**Confidence:** 4

**Summary:**

This paper introduces extra information from an external database during test-time adaptation via retrieval. The method is designed as a plug-and-play module for various  TTA methods.

**Strengths:**

- This paper is well-organized.

**Weaknesses:**

- This paper does not discuss the difference between some highly relevant papers.
- No analysis of the influence of different external database choices.
- Some arguments in the paper need further justification.

**Questions:**

- Retrieval-augmented TTA is not something new. It is a pity that the paper ignores some highly relevant papers with similar ideas. Please see papers [1,2]. Considering these existing works, the reviewer thinks the contribution of this paper needs further justification.


- In lines 074-076, the paper claims "simATTA (Gui et al., 2024) was the first work that emphasized the necessity of extra information for TTA". If we consider the existence of [1], should we modify the argument?

- In lines 076-078, the paper claims "However, this training-based approach accompanies high computational overhead and likely incurs catastrophic forgetting that undermines the knowledge from pre-training data." Can we directly reset the parameters of the TTA model to avoid catastrophic forgetting, as many TTA works, like TPT?

- In Table 1, the improvement of REACT is relatively trivial compared to that in Table 2. What is the reason behind this? The nature of the used external database or something others.

- The reviewer does not find the discussion about different external retrieval sources/databases. Any discussion on this point would be great.

[1] Zancato et al. Train/Test-Time Adaptation with Retrieval. https://arxiv.org/pdf/2303.14333

[2] Lee et al. RA-TTA: Retrieval-Augmented Test-Time Adaptation for Vision-Language Models. ICLR 2025. https://openreview.net/pdf?id=V3zobHnS61.

---

### Official Review · Reviewer_8oXY · 2025-10-30

**Soundness:** 3
**Presentation:** 3
**Contribution:** 2
**Rating:** 4
**Confidence:** 3

**Summary:**

This paper proposes REACT, a plug-and-play retrieval-augmented active test-time adaptation framework for pre-trained vision-language models (VLMs) such as CLIP. REACT aims to improve robustness against distribution shift by selectively querying human labels for uncertain samples and correcting future predictions via retrieval from an incrementally constructed labeled database. The approach is training-free and broadly compatible with diverse TTA pipelines. The authors empirically demonstrate REACT’s effectiveness on both out-of-distribution and cross-domain benchmarks, showcasing consistent improvements over strong TTA baselines.

**Strengths:**

1. Generalizable Plug-and-play Design: The method is training-free and broadly compatible with existing TTA frameworks.

2. Clear Motivation and Context: The paper provides a thorough explanantion of how pseudo labels can fail under distribution shift, and why selective active querying can alleviate the issue. Figures were used to effectively illustrate these concepts, showing scenarios where retrieval-based correction is likely to help or fail.

3. Comprehensive Experiments and Solid Results: The method is rigorously evaluated on multiple VLM architectures (ResNet50, ViT-B/16, BLIP-2, SigLIP), broad OOD and cross-domain benchmarks, and compared to a range of baselines including TTA, active learning, and prompt learning methods. Additionally, the paper provides visual analysis of decision boundaries and correction effects, runtime inference cost analysis and hyperparameter sensitivity tests.

**Weaknesses:**

1. Limited Empirical Scope in Failure Cases: The paper focuses on positive results and does not sufficiently analyze or visualize where REACT fails across diverse domain shifts, class imbalance, or when labeled examples are misleading. It would benefit from explicit breakdowns and more detailed diagnostic visualizations of such cases.

2. Lack of Differentiation from Training-based Methods: The comparison with training-based methods such as simATTA and prompt learning (e.g., CoOp) is made, but it seems counter-intuitive that REACT’s consistent outperforms those methods, and the underlying reason is not deeply explored. Could the benefits come from specific tuning or implementation details?

3. Heuristic Nature of Hyperparameters: The framework’s main metrics, consistency and reliability, relies on two thresholds ($\tau_u$, $\tau_r$). Ablations show that the method is somewhat robust, but there is little justification for the choice of these thresholds. The method could benefit from attempts to learn or adapt the hyperparameters for deployment.

**Questions:**

1. How does REACT scale with the size of the labeled set, particularly when test streams are long?

2. What is the impact if the oracle labeling budget is used up well before the stream ends? Are there mechanisms for budget allocation over time, like prioritizing late-stage samples?

3. In practice, how should $\tau_u$ and $\tau_r$ be set, and can the system adapt or learn these thresholds in an online manner rather than choosing a fixed value?

---

### Official Review · Reviewer_FboZ · 2025-10-31

**Soundness:** 1
**Presentation:** 3
**Contribution:** 2
**Rating:** 2
**Confidence:** 4

**Summary:**

This paper introduces REACT, a plug-and-play framework for active test-time adaptation of vision-language models (VLMs) that reduces reliance on noisy pseudo-labels by selectively querying human experts for ground-truth labels on uncertain samples. Instead of retraining the model, REACT builds a growing database of labeled examples and uses retrieval-augmented correction—guided by consistency and reliability criteria between neighboring labeled samples—to improve predictions on future test inputs. The approach significantly boosts performance across multiple VLMs and benchmarks (like ImageNet variants and cross-domain datasets) with negligible computational overhead, offering an efficient, training-free alternative to existing test-time adaptation methods.

**Strengths:**

- REACT introduces a novel, training-free framework for active test-time adaptation of vision-language models that significantly improves robustness under distribution shifts without requiring any model retraining.
- The plug-and-play design seamlessly integrates with any existing TTA method (e.g., TPT, TDA), delivering consistent performance gains across diverse benchmarks (OOD and cross-domain) with minimal computational overhead (<1 ms per sample).
- The concept of actively seeking external knowledge is intriguing, reminiscent of how humans approach and solve complex questions.

**Weaknesses:**

- The framework relies on access to ground-truth labels during test time, which fundamentally shifts the problem from zero-shot test-time adaptation to a few-shot setting, undermining claims of "zero-shot" generalization and making comparisons with purely unsupervised TTA methods unfair.
- The retrieval-augmented correction mechanism closely resembles memory-augmented or nearest-neighbor approaches like Tip-Adapter, offering incremental improvements without introducing novel architectural or representational innovations.
- The requirement for a labeled buffer built via active querying introduces practical deployment barriers in real-world scenarios where human-in-the-loop annotation is costly, slow, or infeasible, limiting real-world applicability despite claims of “plug-and-play” efficiency.
- The evaluation focuses on benchmark datasets where class sets are predefined and closed, failing to address the more challenging case of truly unseen or open-ended classes — the very problem TTA aims to solve.
- The paper conflates active learning with test-time adaptation; its core contribution is better framed as a streaming, semi-supervised refinement method rather than a true adaptation of the model’s internal representation under distribution shift.

**Questions:**

See Weaknesses.

---

### Note · Authors · 2025-11-12

**Comment:**

After careful consideration of the reviewers' comments, we have decided to withdraw our paper from further consideration at this time.

**Withdrawal Confirmation:**

I have read and agree with the venue's withdrawal policy on behalf of myself and my co-authors.